# GEO-3DGS: MULTI-VIEW GEOMETRY CONSISTENCY FOR 3D GAUSSIAN SPLATTING AND SURFACE RECONSTRUCTION

## ABSTRACT

Recently, the emergence of 3D Gaussian Splatting (3DGS) has made real-time and high-quality rendering possible. However, it is still challenging for 3DGS to reconstruct accurate geometry surfaces and achieve higher-quality rendering. To address these challenges, we propose to leverage multi-view geometry consistency for 3DGS and surface reconstruction. We reveal that there exists multi-view geometry inconsistency in 3DGS, preventing 3DGS from achieving higher-quality rendering and accurate surface reconstruction. To mitigate the geometry inconsistency, we first develop a multi-view photometric consistency regularization to constrain the rendered depth of 3DGS, which helps establish more stable and consistent 3D Gaussians to facilitate both rendering and surface reconstruction. To reconstruct geometry surfaces from 3DGS, we introduce a neural Signed Distance Function (SDF) field to represent continuous geometries of 3DGS. Then, we propose a geometry consistency-based SDF learning strategy, which leverages multi-view geometry consistency cues from 3DGS to efficiently optimize the SDF field for surface reconstruction. Extensive experiments on various datasets demonstrate that our method achieves both high-quality rendering and accurate surface reconstruction while keeping a good efficiency. Our code will be released upon publication.

## 1 INTRODUCTION

With the development of neural rendering Mildenhall et al. (2020); Zhang et al. (2020) and implicit fields Park et al. (2019); Mescheder et al. (2019), realizing Novel View Synthesis (NVS) and surface reconstruction in a unified framework has become an important topic in computer vision and graphics. Neural Radiance Field (NeRF) Mildenhall et al. (2020); Barron et al. (2021); Fridovich-Keil et al. (2022); Müller et al. (2022) methods represent 3D scenes as a color field and density field and leverage neural volume rendering Curless & Levoy (1996) to optimize these fields. After training, these methods can not only synthesize novel views but also reconstruct geometry surfaces from the density field. In fact, the density field is an approximation of actual geometry, thus usually leading to noisy and biased geometry reconstructions. Some pioneering methods Wang et al. (2021); Yariv et al. (2021) introduce Signed Distance Function (SDF) fields to represent actual geometry and derive an SDF-to-density transformation for neural volume rendering. These methods have demonstrated impressive surface reconstruction. However, the massive field query used in neural volume rendering severely compromises real-time rendering capabilities.

Recently, the emergence of 3D Gaussian Splatting (3DGS) Kerbl et al. (2023) has made real-time and high-quality rendering possible. This can be attributed to the fact that 3DGS represents 3D scenes as explicit and discrete 3D Gaussian primitives and performs $\alpha$-blending with differentiable rasterization. However, the above two characteristics also pose challenges to reconstruct accurate geometry surfaces and achieve higher-quality rendering due to the following reasons: 1) Discrete geometry representations typically result in unstable optimization of Gaussian point distribution and incomplete geometry reconstructions. As shown in the top of Figure 1(a), the Gaussian points of 3DGS are almost distributed over the object while are very sparse in textureless regions. 2) Point-based $\alpha$-blending in differentiable rasterization causes most learned 3D Gaussians to deviate from true geometry surfaces. This makes the reconstructed surface from rendered depths of 3DGS noisy

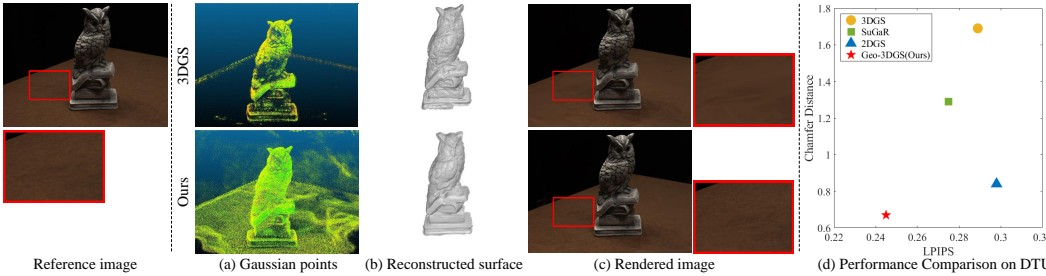

Figure 1: **Motivation.** The 3D Gaussians of 3DGS Kerbl et al. (2023) are discrete and incomplete. The $\alpha$-blending causes most learned 3D Gaussians to deviate from true geometry surfaces, resulting in noisy and biased surface reconstructions. The defective geometry of 3DGS also degrades the rendering performance. In contrast, our learned 3D Gaussians are structured and more completed. This facilitates high-quality surface reconstruction and rendering. Therefore, our method outperforms previous SOTA methods by a large margin on DTU dataset Aanæs et al. (2016).

and biased, as shown in the top of Figure 1(b). As a result, the defective geometry prevents 3DGS from achieving higher-quality rendering, such as the obvious blurry effects shown in the top of Figure 1(c).

To address these issues, we propose Geo-3DGS, a novel approach which leverages multi-view geometry consistency for 3DGS and surface reconstruction. We reveal that the rendered depths from 3DGS are *geometrically inconsistent* across multiple views. In Figure 2(b), we show the rendered depth of 3DGS and check its geometry consistency with the rendered depths of neighboring viewpoints (More geometry consistency check details in Sec. 3.1). The consistency map indicates that the geometry of 3DGS is not consistent across multiple views. This leads to instability of 3D Gaussians and multi-view geometry bias, degrading the 3DGS rendering and making accurate surface reconstruction difficult. To this end, we first develop a multi-view photometric consistency regularization to explicitly constrain the rendered depth of 3DGS. This helps establish more stable 3D Gaussians across multiple views and mitigate the geometry inconsistency in 3DGS, as shown in the bottom of Figure 1(a) and Figure 2(c). Moreover, to reconstruct geometry surfaces from 3DGS with these geometry-aware Gaussians, we introduce a neural SDF field to represent continuous geometries of 3DGS. Then, we propose a geometry consistency-based SDF learning strategy, which leverages multi-view geometry consistency cues derived from 3DGS to efficiently optimize the SDF field for surface reconstruction. In this way, our proposed two consistency strategies promote multi-view geometry consistency for 3DGS and surface reconstruction, thus improving both 3DGS rendering and geometry reconstruction, as shown in the bottom of Figure 1(b) and (c) and Figure 1(d).

In summary, our main contributions are as follows:

- We reveal that there exists multi-view geometry inconsistency in 3DGS, which prevents 3DGS from achieving higher-quality rendering and reconstructing accurate geometry surfaces (Sec. 3.1).
- We develop a multi-view photometric consistency regularization to explicitly constrain the geometry of 3DGS, which helps establish more stable and consistent 3D Gaussians to facilitate both rendering and reconstruction (Sec. 3.2).
- We propose a geometry consistency-based SDF learning strategy for 3DGS, which leverages multi-view geometry consistency cues derived from 3DGS to efficiently optimize SDF fields for surface reconstruction (Sec. 3.3).
- Extensive experiments on indoor and large-scale outdoor datasets verify that our method is capable of achieving high-quality rendering and reconstruction in a unified framework for 3DGS while keeping a good efficiency (Sec. 4).

## 2 RELATED WORK

**Traditional scene reconstruction and rendering.** Traditional image-based rendering methods Chaurasia et al. (2013); Hedman et al. (2018); Riegler & Koltun (2020); Kopanas et al. (2021)

(a) Reference image and GT depth    (b) Rendered depth and consistency map of 3DGS    (c) Our rendered depth and consistency map

Figure 2: **Multi-view geometry inconsistency in 3DGS.** Although the rendered depth of 3DGS looks reasonable, it is *inconsistent* with depth maps of its neighboring views, as indicated by its consistency map (black/white means inconsistency/consistency). In contrast, our method renders more consistent depth maps.

usually reconstruct 3D proxy geometry to guide the synthesis. The 3D proxy geometry is usually obtained by Structure-from-Motion (SfM) Snavely et al. (2006); Schonberger & Frahm (2016), Multi-View Stereo (MVS) Schönberger et al. (2016); Xu & Tao (2019; 2020a); Xu et al. (2022); Xu & Tao (2020b); Ren et al. (2023) and surface reconstruction Kazhdan & Hoppe (2013); Labatut et al. (2007); Curless & Levoy (1996) step by step. With the 3D proxy geometry, these methods reproject and blend input images into novel viewpoints. These methods have demonstrated that better geometry can help NVS. However, they heavily rely on geometry for reprojection, making their rendering results susceptible to noise or geometry loss. Our method represents the scene geometry with 3D Gaussians and improves NVS with better geometry guidance.

**Neural rendering and implicit surface reconstruction.** Neural implicit fields have been widely used to represent 3D scenes. NeRF Mildenhall et al. (2020), which uses the Multi-Layer Perceptron (MLP) to represent radiance fields, has inspired numerous works to use neural implicit fields for NVS and surface reconstruction. On one hand, many works Fridovich-Keil et al. (2022); Sun et al. (2021); Müller et al. (2022); Chan et al. (2022) are proposed to use efficient data structures to improve training and rendering speed. On the other hand, some methods Wang et al. (2021); Yariv et al. (2021); Fu et al. (2022); Darmon et al. (2021); Wang et al. (2022); Yu et al. (2022); Li et al. (2023); Su et al. (2024) introduce SDF to represent the true geometry and improve the SDF learning. However, these methods still cannot achieve real-time rendering. In this work, we adopt 3DGS for radiance fields and neural implicit SDF fields for geometry fields, combining the advantages of both.

**3D Gaussian Splatting (3DGS) and surface reconstruction.** 3DGS Kerbl et al. (2023) has achieved impressive NVS performance in terms of both rendering quality and speed. Subsequently, many follow-ups Yu et al. (2023); Lu et al. (2023); Cheng et al. (2024) are proposed to tackle aliasing issues, Gaussian redundancy and large-scale scene rendering. Meanwhile, some concurrent works Guédon & Lepetit (2023); Chen et al. (2023); Yu et al. (2024a); Lyu et al. (2024) explore surface reconstruction for 3DGS. SuGaR Guédon & Lepetit (2023) adds a regularization term to align Gaussians with the scene surface better and then uses Poisson surface reconstruction Kazhdan et al. (2006). 2DGS Huang et al. (2024) uses 2D Gaussians to reconstruct geometrically accurate radiance fields and fuses rendered depth maps to extract meshes. GOF Yu et al. (2024b) presents Gaussian opacity fields to identify the level set of 3D Gaussians for mesh extraction. Unlike these methods, our method proposes a multi-view photometric consistency regularization to mitigate the inconsistency in 3DGS and designs a geometry consistency-based SDF learning strategy for surface reconstruction, achieving a better balance between NVS, surface reconstruction and efficiency.

## 3 METHOD

Given posed multi-view images of a scene, we aim to reconstruct the 3D scene with photorealistic NVS and accurate surface reconstruction. We represent the radiance field with 3D Gaussians and the geometry field with an SDF, synthesizing novel views through splatting rendering of 3D Gaussians and extracting the surface using the zero-level set of SDF. During training, our goal is to optimize the 3D Gaussians and SDF simultaneously. In this section, we first reveal the multi-view geometry inconsistency in 3DGS which prevents 3DGS from achieving high-quality rendering and surface reconstruction (Sec. 3.1). Then we develop a multi-view photometric consistency regularization to mitigate this inconsistency (Sec. 3.2). Finally, we design a geometry consistency-based SDF learning for surface reconstruction (Sec. 3.3). An overview of our approach is shown in Figure 3.

Figure 3: **Overview of Geo-3DGS.** For a given viewpoint, we render its image and depth map from 3D Gaussians. Our proposed multi-view photometric consistency regularization leverages the multi-view patch matching technique to optimize the rendered depth through neighboring images, thus mitigating the multi-view geometry inconsistency in 3DGS. On this basis, our geometry consistency-based SDF learning leverages multi-view geometric consistency cues to facilitate efficient SDF learning. Moreover, our geometry assisted densification introduces a more complete point cloud fused from all rendered depth maps to guide the optimization of 3D Gaussians.

## 3.1 MULTI-VIEW GEOMETRY INCONSISTENCY IN 3DGS

In the process of splatting rendering, there exists multi-view geometry inconsistency in 3DGS which poses challenges to achieve high-fidelity geometry reconstruction and NVS.

**3D Gaussian Splatting (3DGS).** 3DGS Kerbl et al. (2023) represents 3D scenes by 3D Gaussian primitives and performs $\alpha$-blending with differentiable rasterization to render images. Each Gaussian primitive is defined with its mean vector $\boldsymbol{\mu} \in \mathbb{R}^{3 \times 1}$, covariance matrix $\boldsymbol{\Sigma} \in \mathbb{R}^{3 \times 3}$, color $\mathbf{c} \in \mathbb{R}^{3 \times 1}$ and opacity $\alpha \in \mathbb{R}$. For a 3D spatial point $\mathbf{x}$, the 3D Gaussian can be queried as:

$$\mathcal{G}(\mathbf{x}) = \exp(-\frac{1}{2}(\mathbf{x} - \boldsymbol{\mu})^T \boldsymbol{\Sigma}^{-1}(\mathbf{x} - \boldsymbol{\mu})). \tag{1}$$

To render an image or a depth at a given viewpoint, 3DGS projects 3D Gaussians into the image plane and employs $\alpha$-blending to calculate the color/depth of a pixel $\mathbf{p}$ as:

$$\mathbf{c}(\mathbf{p}) = \sum_{i=1}^{N} w_i \mathbf{c}_i, \quad d(\mathbf{p}) = \frac{\sum_{i=1}^{N} w_i d_i}{\sum_{i=1}^{N} w_i}, \quad w_i = \sigma_i \prod_{j=1}^{i-1}(1 - \sigma_j), \tag{2}$$

where $\sigma_i$ is computed by evaluating a projected 2D Gaussian from $\mathcal{G}_i$ in $\mathbf{p}$ multiplied with $\alpha_i$, $d_i$ refers to the projected depth of the $i$-th Gaussian under the current viewpoint and $N$ denotes the number of ordered Gaussians overlapping the pixel $\mathbf{p}$. Since the accumulated alpha $A(\mathbf{p}) = \sum_{i=1}^{N} w_i$ may not reach saturation, it is important to normalize the depth with the accumulated alpha when rendering depth. 3DGS exploits a tile-based rasterizer to achieve real-time rendering speed and optimizes Gaussian primitives with an adaptive density control strategy.

**Multi-view geometry consistency check.** 3DGS represents scene geometry as a series of discrete 3D Gaussians. To investigate multi-view geometry consistency for 3DGS, we leverage rendered depth maps of 3DGS to check multi-view geometry consistency Schönberger et al. (2016); Xu & Tao (2019; 2020a); Xu et al. (2022); Ren et al. (2023). To achieve this, given the current rendered depth map $\mathbf{D}_0$, we also render its neighboring depth maps $\{\mathbf{D}_k\}_{k=1}^{K}$ according to its neighboring viewpoints, where $K$ denotes the number of neighboring viewpoints. For the depth $\mathbf{D}_0(\mathbf{p})$ of pixel $\mathbf{p}$ in the current viewpoint, its projected pixel in the $k$-th neighboring viewpoint is:

$$\tilde{\mathbf{p}}_{0,k} = P_{0,k}(\mathbf{p}) = \mathbf{K}_k(\mathbf{R}_{0,k}\mathbf{x}(\mathbf{p}) + \mathbf{t}_{0,k}), \quad \mathbf{x}(\mathbf{p}) = \mathbf{K}_0^{-1}\mathbf{D}_0(\mathbf{p})\tilde{\mathbf{p}}, \tag{3}$$

where $P_{0,k}(\cdot)$ denotes the projection operation, $\tilde{\mathbf{p}}$ is the homogeneous coordinate of $\mathbf{p}$, $\mathbf{x}(\mathbf{p})$ is the 3D point of $\mathbf{p}$ in the current camera coordinate, $\mathbf{K}_0$ and $\mathbf{K}_k$ are the intrinsic parameters of the current image and the $k$-th neighboring image, $\mathbf{R}_{0,k}$ and $\mathbf{t}_{0,k}$ are the relative rotation and translation. In this way, we can query the depth information of pixel $\mathbf{p}_{0,k}$ from the $k$-th depth map to obtain $\mathbf{D}_k(\mathbf{p}_{0,k})$,

where $\mathbf{p}_{0,k}$ is the dehomogeneous coordination of $\tilde{\mathbf{p}}_{0,k}$. By reprojecting $\mathbf{p}_{0,k}$ into the current image with $\mathbf{D}_k(\mathbf{p}_{0,k})$, we obtain the relative depth difference and reprojection error of pixel $\mathbf{p}$ as:

$$e_{\text{diff}} = \frac{|z(P_{k,0}(\mathbf{p}_{0,k})) - \mathbf{D}_0(\mathbf{p})|}{\mathbf{D}_0(\mathbf{p})}, \quad e_{\text{reproj}} = ||\frac{P_{k,0}(\mathbf{p}_{0,k})}{z(P_{k,0}(\mathbf{p}_{0,k}))} - \tilde{\mathbf{p}}||_2, \tag{4}$$

where $z(\cdot)$ denotes the depth value for a homogeneous coordinate. When $e_{\text{diff}} < \epsilon_{\text{diff}}$ and $e_{\text{reproj}} < \epsilon_{\text{reproj}}$, the pixel $\mathbf{p}$ is deemed two-view consistent. By checking the two-view consistency of $\mathbf{p}$ with respect to different neighboring viewpoints, if there exist at least $n$-view consistent views, the pixel $\mathbf{p}$ is considered to satisfy multi-view geometric consistency. Using this criterion, we check the multi-view geometry consistency for 3DGS. As shown in Figure 2(b), the rendered depth map of 3DGS is *barely* consistent with its neighboring rendered depth maps. This demonstrates multi-view geometry inconsistency in 3DGS. This inconsistency makes the learned 3D Gaussians unstructured and incomplete, preventing 3DGS from achieving high-quality surface reconstruction and rendering, as shown in the top of Figure 1(b) and (c).

### 3.2 MULTI-VIEW GEOMETRIC CONSISTENCY-BASED 3DGS REGULARIZATION AND DENSIFICATION

The rendered depth of 3DGS describes the geometry of target scenes. To mitigate the multi-view geometry inconsistency in 3DGS, we propose a multi-view photometric consistency regularization to optimize the rendered depth of 3DGS, which encourages the geometry of 3DGS to be geometrically consistent across multiple views. Based on this, we further utilize the geometry-consistent depth cues derived from the 3DGS itself for densification to overcome the under-reconstruction and over-reconstruction problems.

**Multi-view photometric consistency regularization.** Inspired by the PatchMatch MVS Schönberger et al. (2016); Xu & Tao (2019); Xu et al. (2022); Xu & Tao (2020a); Ren et al. (2023), we resort to 3D plane models to constrain the geometry of 3DGS by multi-view photometric consistency. A 3D plane model can be represented by $\pi = [\mathbf{n}^\top, \tilde{d}]^\top$, where $\mathbf{n}$ is normal vector and $\tilde{d}$ is distance from a 3D plane to the origin. In order to define the 3D plane model for pixel $\mathbf{p}$, it is important to compute its normal information. While the normal information can be obtained by $\alpha$-blending of Gaussian directions Jiang et al. (2023), we find that the rendered normal from 3DGS is noisy and inconsistent with the rendered depth. Consequently, this 3D plane model cannot effectively constrain the geometry of 3DGS. Therefore, we approximate its normal vector $\tilde{\mathbf{n}}(\mathbf{p})$ based on the point map computed from the rendered depth map $\mathbf{D}_0$ Jiang et al. (2023); Huang et al. (2024) and calculate its corresponding $\tilde{d}(\mathbf{p})$ as:

$$\tilde{\mathbf{n}}(\mathbf{p}) = \frac{\nabla_x \mathbf{x}(\mathbf{p}) \times \nabla_y \mathbf{x}(\mathbf{p})}{|\nabla_x \mathbf{x}(\mathbf{p}) \times \nabla_y \mathbf{x}(\mathbf{p})|}, \quad \tilde{d}(\mathbf{p}) = \tilde{\mathbf{n}}(\mathbf{p})^\top \mathbf{x}(\mathbf{p}). \tag{5}$$

With the above defined 3D plane model, a small area centered on $\mathbf{x}(\mathbf{p})$ on the 3D plane can be projected into the current image and the $k$-th neighboring image to obtain two small image patches respectively. Then, the image pixel $\mathbf{p}$ in the current image patch $q$ and the image pixel $\mathbf{p}'$ in the neighboring image patch $q_k$ are related by the plane-induced homography $\mathbf{H}_{0,k}$ Hartley & Zisserman (2004):

$$\mathbf{p}' = \mathbf{H}_{0,k}\mathbf{p}, \quad \mathbf{H}_{0,k} = \mathbf{K}_k(\mathbf{R}_{0,k} - \mathbf{t}_{0,k}\frac{\mathbf{n}(\mathbf{p})^\top}{\tilde{d}(\mathbf{p})})\mathbf{K}_0^{-1}. \tag{6}$$

In practice, we first select the image patch $q$ of size $11 \times 11$ centered on pixel $\mathbf{p}$, then leverage Eq. (6) to determine the corresponding neighboring image patch $q_k$. Following Schönberger et al. (2016); Xu & Tao (2019), we utilize the Normalized Cross Correlation (NCC) to measure the photometric consistency between $q$ and $q_k$ as $\text{NCC}(q, q_k)$. To alleviate the influence of occlusions, we choose the best $M$ of the NCC scores calculated for $K$ neighboring images to define the multi-view photometric consistency loss as:

$$\mathcal{L}_{\text{photo}_{\text{3DGS}}} = \frac{\sum_{m=1}^{M} 1 - \text{NCC}(q, q_k')}{M}, \tag{7}$$

where $\{\text{NCC}(q, q_k')\}_{k=1}^{K}$ means the sorted NCC scores, *i.e.*, $\text{NCC}(q, q_1') < \text{NCC}(q, q_2') < \cdots < \text{NCC}(q, q_K')$. The multi-view photometric consistency loss indicates the precision of the rendered

depth of 3DGS to some extent. With this loss, the rendered depth of 3DGS will become more accurate and the geometry of GS can be guided to be more consistent across multiple views.

**Geometry assisted Gaussian densification.** 3DGS initializes Gaussians with sparse point clouds from SfM algorithms Snavely et al. (2006); Schonberger & Frahm (2016), and applies the adaptive Gaussian densification strategy to progressively generate new Gaussians, enhancing the scene representation gradually Kerbl et al. (2023). The adaptive Gaussian densification strategy Kerbl et al. (2023) either clones the small Gaussians to the direction of the positional gradient or splits the large Gaussians into smaller ones. However, since the SfM algorithms can only recover the very basic structure of the scene and usually fail to produce point clouds in low-texture areas, the Gaussians of 3DGS still can not describe the scene accurately after densification through the less-constrained densification strategy Cheng et al. (2024) as illustrated in Figure 1(a). In this case, the under-reconstruction and over-reconstruction problems still pose challenges for 3DGS. To solve this, we resort to leverage the the geometry-consistent cues of 3DGS for densification.

Through our above regularization, the depths rendered from the 3DGS become more geometrically consistent and accurate. These rendered depths can be used to recover the dense point cloud which can represent the structure of the scene more accurately and delicately compared with that of the sparse point clouds recovered by SfM Snavely et al. (2006); Schonberger & Frahm (2016). To this end, after obtaining the rendered depths of all training views, the multi-view geometry consistency check mentioned in Sec. 3.1 is used to filter the unreliable depth, namely, if the depth $\mathbf{D}_0(\mathbf{p})$ is at least $n$-view consistent, it is regarded reliable. To further reduce the noise in the rendered depth, the reliable depth $\mathbf{D}_0(\mathbf{p})$ is averaged with the reprojected depths $\{z(P_{k,0}(\mathbf{p}_{0,k}))\}_{k \in \mathbf{I}_c}$ from its all consistent neighboring views to get $\bar{\mathbf{D}}_0(\mathbf{p})$ as:

$$\bar{\mathbf{D}}_0(\mathbf{p}) = (\mathbf{D}_0(\mathbf{p}) + \sum_{k \in \mathbf{I}_c} z(P_{k,0}(\mathbf{p}_{0,k})))/(|\mathbf{I}_c| + 1). \tag{8}$$

Finally, all $\bar{\mathbf{D}}_0(\mathbf{p})$ are projected to the 3D world space to obtain a dense point cloud. Naive applying this for densification will greatly increase the computational burden of 3DGS, so the downsampled point cloud is used for densification by initializing them as new Gaussians, yielding Gaussians with a more uniform distribution throughout the entire scene to represent the scene well.

### 3.3 GEOMETRY CONSISTENCY-BASED SDF LEARNING FOR 3DGS

As the 3DGS is designed for the NVS task, it does not explicitly model the surface geometry of the scene. Thus, it typically needs to use Poisson reconstruction Kazhdan et al. (2006) or TSDF fusion Curless & Levoy (1996) as a post-processing step to recover the surface mesh Guédon & Lepetit (2023); Huang et al. (2024), which is not efficient and unable to guarantee accuracy of recovered geometry. To this end, we introduce a neural SDF to explicitly model the surface geometry and exploit the geometric information derived from the 3DGS to constrain the learning of the SDF. We propose a two-stage training strategy for efficient SDF learning: the first stage leverages the depth rendered by 3DGS to guide the SDF initialization, and the second stage incorporates the geometry consistency-aware regularization to refine the learned SDF.

**3DGS-driven SDF initialization.** Our SDF is modeled by a geometry network consisting of multi-resolution hash grids Müller et al. (2022) and MLPs with learnable parameters $\theta$. The zero-level set of the SDF defines the surface $\mathcal{S}$ of a scene as $\mathcal{S} = \{\mathbf{x} \in \mathbb{R}^3 | f_\theta(\mathbf{x}) = 0\}$. To leverage the depth $\mathbf{D}_0(\mathbf{p})$ rendered by 3DGS for SDF learning, the volume rendering technique Mildenhall et al. (2020) and SDF-induced density function Wang et al. (2021) are introduced to covert the level sets of SDF to depth:

$$\hat{\mathbf{D}}(\mathbf{p}) = \sum_{i=1}^{N} T_\mathbf{r}^i \alpha_\mathbf{r}^i t_\mathbf{r}^i, \quad T_\mathbf{r}^i = \prod_{j=1}^{i-1} \left(1 - \alpha_\mathbf{r}^j\right), \quad \alpha_\mathbf{r}^i = \max\left(\frac{\Phi_s(f(\mathbf{x}_\mathbf{r}^i)) - \Phi_s(f(\mathbf{x}_\mathbf{r}^{i+1}))}{\Phi_s(f(\mathbf{x}_\mathbf{r}^i))}, 0\right), \tag{9}$$

where $\Phi_s$ denotes the Sigmoid function, $T_\mathbf{r}^i$, $t_\mathbf{r}^i$ and $\alpha_\mathbf{r}^i$ denote the transmittance, projected depth and opacity of the $i$-th point along ray $\mathbf{r}$ that is corresponded to the sampled pixel $\mathbf{p}$, respectively. Note that to locate the near-surface points more efficiently during training, we additionally sample $N_{gs}$ samples around $\mathbf{x}(\mathbf{p})$ after obtaining samples by the hierarchical sampling algorithm Wang et al. (2021) to form the final $N$ samples along the ray $\mathbf{r}$.

To initialize SDF more efficiently, the depth $\mathbf{D}_0(\mathbf{p})$ rendered by 3DGS is used to constrain the learning of both all level sets and the zero-level set of SDF as follows:

$$\mathcal{L}_{\text{depth}} = \frac{1}{S}\sum_{s=1}^{S}\left|\hat{\mathbf{D}}(\mathbf{p}^s) - \mathbf{D}_0(\mathbf{p}^s)\right|, \quad \mathcal{L}_{\text{zero}} = \frac{1}{S}\sum_{s=1}^{S}(|f_\theta(\mathbf{x}(\mathbf{p}^s))| - \epsilon), \tag{10}$$

where $S$ denotes the number of sampled pixels in a batch during training. Since the priors obtained from 3DGS are not accurate enough, a small scalar $\epsilon$ is used in $\mathcal{L}_{\text{zero}}$ and is set as 0.001. In addition, a regularization loss $\mathcal{L}_{\text{reg}}$ is further introduced to regularize the noise in the priors, which is given by:

$$\mathcal{L}_{\text{reg}} = \frac{1}{S}\sum_{s=1}^{S}(1-\sum_{i=1}^{N}(w_{\text{reg}}^{s,i}T_{\mathbf{r}}^{s,i}\alpha_{\mathbf{r}}^{s,i})+\sum_{i=1}^{N}((1-w_{\text{reg}}^{s,i})T_{\mathbf{r}}^{s,i}\alpha_{\mathbf{r}}^{s,i})), w_{\text{reg}}^{s,i} = [|\hat{\mathbf{D}}(\mathbf{p^s})-t_{\mathbf{r}}^{s,i}| < \frac{\delta_{\mathbf{r}}^s}{N}], \tag{11}$$

where $[\cdot]$ is Iverson bracket, $\delta_{\mathbf{r}}^s$ is the distance between the near plane and far plane.

**Geometry-consistent SDF refinement.** After initialization, the basic structure of the scene is learned. To promote the geometric consistency of learned SDF and recover the delicate details of the scene, the geometric consistency constraint is introduced in the SDF refinement stage.

Instead of using priors to constrain the learning of zero-level sets of SDF, the multi-view photometric consistency loss Fu et al. (2022) is used to optimize the learned SDF. Following Fu et al. (2022), we locate the zero-level set of SDF to obtain the surface point $\hat{\mathbf{x}}$. Based on the depth $\hat{d}$ of $\hat{\mathbf{x}}$ and normal $\hat{\mathbf{n}}$ computed with automatic differentiation of geometry network $f_\theta$ at $\hat{\mathbf{x}}$, by adopting Eq. (5) - (7), the multi-view photometric consistency loss $\mathcal{L}_{\text{photo}_{\text{SDF}}}$ for the optimization of SDF is obtained. In this stage, the depth rendered by the 3DGS are still utilized to constrain the learning of all level sets of SDFs.

It is worth noting that unlike previous implicit surface reconstruction methods Yariv et al. (2021); Wang et al. (2021) that require color fields to drive SDF learning, our Geo-3DGS utilizes geometric consistency information obtained from 3DGS to guide SDF learning. This not only ensures accuracy of recovered geometry and improves SDF training speed, but also achieves high fidelity real-time rendering based on 3DGS.

## 3.4 Loss Function

The losses used for 3DGS and neural SDF constitute the overall loss for Geo-3DGS, namely, $\mathcal{L}_{\text{Geo-3DGS}} = \mathcal{L}_{\text{3DGS}} + \mathcal{L}_{\text{SDF}}$. For 3DGS, in addition to using the proposed multi-view photometric consistency loss $\mathcal{L}_{\text{photo}_{\text{3DGS}}}$ to handle the geometric inconsistency problem in 3DGS, the depth-normal consistency loss $\mathcal{L}_{\text{normal}}$ and depth smoothness loss $\mathcal{L}_{\text{smooth}}$ are incorporated to enforce the consistency between the rendered depth and normal of 3DGS and the smoothness of the rendered depth.

$$\mathcal{L}_{\text{normal}} = \frac{1}{N_0}\sum\left\|1-\tilde{\mathbf{n}}^T\bar{\mathbf{n}}\right\|_1, \quad \mathcal{L}_{\text{smooth}} = \frac{1}{N_0}\sum(e^{-|\nabla_x\mathbf{I}|}\nabla_x\mathbf{D}_0 + e^{-|\nabla_y\mathbf{I}|}\nabla_y\mathbf{D}_0), \tag{12}$$

where $\bar{\mathbf{n}}$ denotes the normal rendered by 3DGS and $N_0$ is number of pixel in the image. Combine the color loss $\mathcal{L}_{\text{color}}$ and D-SSIM term $\mathcal{L}_{\text{D-SSIM}}$ with above losses, the overall loss for the 3DGS is:

$$\mathcal{L}_{\text{3DGS}} = (1 - \lambda_1)\mathcal{L}_{\text{color}} + \lambda_1\mathcal{L}_{\text{D-SSIM}} + \lambda_2\mathcal{L}_{\text{photo}_{\text{3DGS}}} + \lambda_3\mathcal{L}_{\text{normal}} + \lambda_4\mathcal{L}_{\text{smooth}}. \tag{13}$$

As for the training of the neural SDF, except for the losses mentioned in Sec. 3.3, the Eikonal term $\mathcal{L}_{\text{eik}}$ Gropp et al. (2020) is applied to regularize the neural SDF. Then, the loss used for the SDF learning is:

$$\mathcal{L}_{\text{SDF}} = \lambda_5\mathcal{L}_{\text{depth}} + \lambda_6\mathcal{L}_{\text{zero}} + \lambda_7\mathcal{L}_{\text{reg}} + \lambda_8\mathcal{L}_{\text{photo}_{\text{SDF}}} + \lambda_9\mathcal{L}_{\text{eik}}, \tag{14}$$

where $\{\lambda_i\}_{i=1}^9$ are weights to balance different loss terms, the $\mathcal{L}_{\text{zero}}$ and $\mathcal{L}_{\text{reg}}$ are only used in the SDF initialization stage, while the $\mathcal{L}_{\text{photo}_{\text{SDF}}}$ is only activated in the SDF refinement stage.

Table 1: **Quantitative reconstruction comparison on DTU.**

| | 24 | 37 | 40 | 55 | 63 | 65 | 69 | 83 | 97 | 105 | 106 | 110 | 114 | 118 | 122 | Mean | Time |
|---|---|---|---|---|---|---|---|---|---|---|---|---|---|---|---|---|---|
| NeRF Mildenhall et al. (2020) | 1.90 | 1.60 | 1.85 | 0.58 | 2.28 | 1.27 | 1.47 | 1.67 | 2.05 | 1.07 | 0.88 | 2.53 | 1.06 | 1.15 | 0.96 | 1.49 | > 12h |
| VolSDF Yariv et al. (2021) | 1.14 | 1.26 | 0.81 | 0.49 | 1.25 | 0.70 | 0.72 | 1.29 | 1.18 | 0.70 | 0.66 | 1.08 | 0.42 | 0.61 | 0.55 | 0.86 | > 12h |
| NeuS Wang et al. (2021) | 1.00 | 1.37 | 0.93 | 0.43 | 1.10 | **0.65** | 0.57 | 1.48 | 1.09 | 0.83 | **0.52** | 1.20 | **0.35** | **0.49** | 0.54 | 0.84 | > 12h |
| 3DGS Kerbl et al. (2023) | 1.85 | 1.50 | 1.63 | 0.95 | 2.94 | 1.85 | 1.50 | 1.98 | 2.11 | 1.41 | 1.66 | 2.19 | 1.26 | 1.18 | 1.35 | 1.69 | **24m** |
| SuGaR Guédon & Lepetit (2023) | 1.59 | 1.13 | 1.10 | 0.57 | 1.77 | 1.54 | 1.15 | 1.70 | 1.83 | 1.26 | 1.06 | 1.74 | 1.02 | 1.07 | 0.87 | 1.29 | > 3h |
| 2DGS Huang et al. (2024) | 0.51 | 0.86 | **0.40** | **0.39** | 1.17 | 1.12 | 0.87 | 1.29 | 1.26 | 0.75 | 0.70 | 1.65 | 0.44 | 0.74 | **0.50** | 0.84 | 45m |
| Ours (Geo-3DGS) | **0.48** | **0.77** | 0.49 | 0.48 | **0.93** | 0.80 | **0.54** | **1.09** | **1.04** | **0.61** | 0.61 | **0.54** | 0.38 | 0.59 | 0.51 | **0.66** | 45m |

## 4 EXPERIMENTS

### 4.1 EXPERIMENTAL SETTING

**Datasets.** We evaluate our Geo-3DGS performance on DTU Aanæs et al. (2016) and Tanks and Temples Knapitsch et al. (2017) datasets. The DTU dataset contains 15 scenes with images of resolution $1600 \times 1200$. We utilize COLMAP Schonberger & Frahm (2016) to calculate a sparse point cloud for each scene and retain the original resolution for training. For the Tanks and Temples dataset, we conduct comparison on 6 scenes and adopt down-sampled images with a resolution of $960 \times 540$ for efficiency. Additionally, we use the evaluation mode defined in 3DGS to split the available images of each scene into training and test set. We reconstruct mesh based on the training split and compare the rendering performance on the test split. For a fair comparison, we employ the same dataset setting to retrain the baseline methods based on 3DGS.

**Baselines.** We compare our method with SOTA implicit methods (NeRF Mildenhall et al. (2020), VolSDF Yariv et al. (2021), NeuS Wang et al. (2021), Geo-Neus Fu et al. (2022) and Neuralangelo Li et al. (2023)) and explicit methods (3DGS Kerbl et al. (2023), SuGaR Guédon & Lepetit (2023), 2DGS Huang et al. (2024) and Scaffold-GS Lu et al. (2023)). Note that, these implicit methods input all images for each scene to reconstruct surface meshes. Thus, we do not compare our NVS performance with these methods.

**Implementation details.** We build our Geo-3DGS upon the open-source 3DGS code and adopt custom CUDA kernels to output images and depth maps for our presented multi-view geometry consistency learning. For the mesh extraction from 3DGS and 2DGS, we also adopt the TSDF used in 2DGS Huang et al. (2024), with setting the voxel size to 0.004 and the truncated threshold to 0.02. We modify the renderer of 3DGS to generate depth maps. All experiments are conducted on an NVIDIA 3090 GPU.

### 4.2 COMPARISONS

**Surface reconstruction.** We first compare the reconstruction of our method and SOTA methods on DTU dataset with Charmfer distance. Table 1 reports the quantitative results. Our method outperforms all implicit methods and explicit methods. Although implicit methods input all images to learn surfaces, our method still surpasses them. For explicit methods, the geometrically inconsistent depth maps from 3DGS result in poorer surface reconstruction. The improvement of SuGaR is limited by its reliance on rendered depth maps to align Gaussians with surfaces. 2DGS alleviates multi-view inconsistency using 2D Gaussians but lacks explicit geometry constraints. In contrast, our explicit multi-view geometry consistency constraints enable high-quality geometry recovery. Table 2 further shows the comparisons on challenging large-scale Tanks and Temples dataset with the $F_1$ score. Our method achieves competitive performance and better efficiency compared with implicit methods. Notably, our method outperforms concurrent explicit methods by a large margin. Qualitative results in Figure 4 show that our method yields more complete and consistent surface quality than others. Moreover, as for training time, our method is much faster than implicit methods and is competitive with explicit methods thanks to our efficient SDF learning with the geometric cues from 3DGS.

**Novel view synthesis.** We compare the NVS performance of our method and baselines on DTU and Tanks and Temples datasets. Table 3 lists the quantitative results. Although SuGaR and 2DGS improve the surface reconstruction for 3DGS, their rendering performance degrades. Although

Table 2: **Quantitative reconstruction comparison on Tanks and Temples.**

|  | Barn | Cat. | Court. | Ign. | Meet. | Truck | Mean | Time |
|---|---|---|---|---|---|---|---|---|
| NeuS Wang et al. (2021) | 0.29 | 0.29 | 0.17 | 0.83 | 0.24 | 0.45 | 0.38 | > 24h |
| Geo-Neus Fu et al. (2022) | 0.33 | 0.26 | 0.12 | 0.72 | 0.20 | 0.45 | 0.35 | > 24h |
| Neuralangelo Li et al. (2023) | **0.70** | **0.36** | **0.28** | **0.89** | **0.32** | **0.48** | **0.50** | > 24h |
| 3DGS Kerbl et al. (2023) | 0.06 | 0.03 | 0.04 | 0.05 | 0.06 | 0.08 | 0.05 | **17m** |
| SuGaR Guédon & Lepetit (2023) | 0.04 | 0.07 | 0.03 | 0.08 | 0.09 | 0.14 | 0.07 | > 3h |
| 2DGS Huang et al. (2024) | 0.34 | 0.20 | 0.15 | 0.33 | 0.13 | 0.31 | 0.24 | 28m |
| Ours (Geo-3DGS) | 0.44 | 0.30 | 0.20 | 0.57 | 0.22 | 0.42 | 0.36 | 48m |

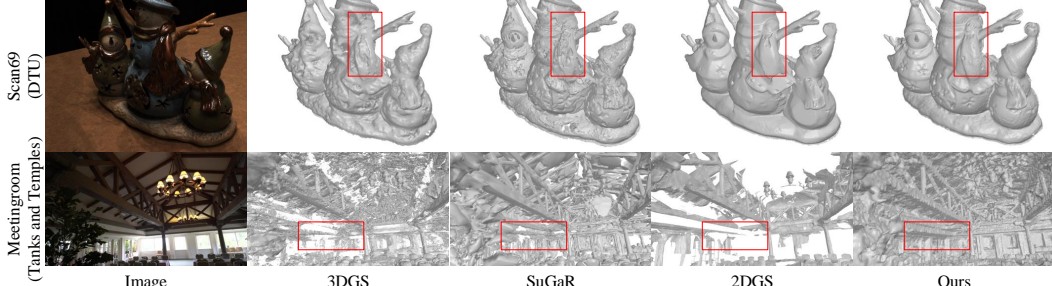

Figure 4: **Qualitative surface reconstruction comparison.**

Scaffold-GS uses anchor points to distribute local Gaussians, these anchor points are not geometry-aware. Therefore, its rendering performance is impaired sometimes. In contrast, since our method leverages multi-view geometry consistency to better guide the distribution of 3D Gaussians in the early training phase, our NVS achieves better rendering than 3DGS. Especially, our method significantly improves the LPIPS, demonstrating our higher-quality rendering. The qualitative results in Figure 5 show that our method can render detailed textures that other methods cannot.

## 4.3 ABLATION STUDY

To evaluate the effect of the core modules in our proposed Geo-3DGS, we conduct an ablation study on Tanks and Temples dataset. 3DGS combined with the 3DGS-driven SDF initialization is adopted as our baseline. Different modules are progressively added to the baseline to investigate their efficacy. The quantitative results are reported in Table 4. See Appendix for qualitative results.

**Geometry-consistent SDF refinement (GCR).** Due to the lack of geometry-consistency constraints, although Baseline can maintain the rendering performance of 3DGS, its reconstruction is very poor. By adding our geometry-consistent SDF refinement, Model-A greatly improves reconstruction performance while maintaining good rendering results.

Table 3: **Quantitative rendering comparison on DTU and Tanks and Temples.**

|  | DTU | | | Tanks and Temples | | |
|---|---|---|---|---|---|---|
|  | PSNR ↑ | SSIM ↑ | LPIPS ↓ | PSNR ↑ | SSIM ↑ | LPIPS ↓ |
| 3DGS Kerbl et al. (2023) | **28.53** | **0.879** | 0.289 | 25.07 | 0.851 | 0.179 |
| SuGaR Guédon & Lepetit (2023) | 27.65 | 0.856 | 0.275 | 23.06 | 0.805 | 0.226 |
| Scaffold-GS Lu et al. (2023) | 28.52 | 0.870 | 0.321 | **25.59** | **0.858** | 0.174 |
| 2DGS Huang et al. (2024) | 27.94 | 0.873 | 0.298 | 24.10 | 0.829 | 0.216 |
| Ours (Geo-3DGS) | 28.22 | 0.876 | **0.244** | 24.93 | 0.854 | **0.170** |

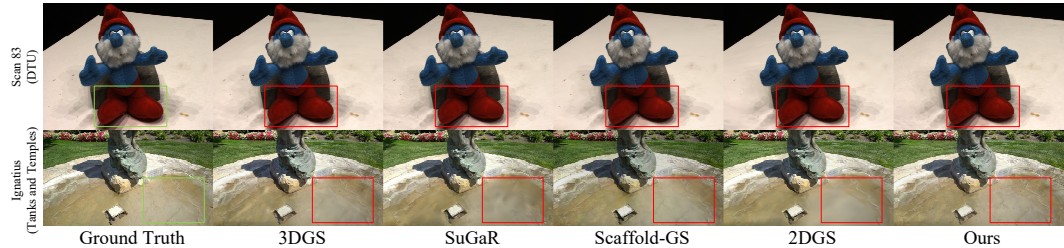

Figure 5: **Qualitative rendering comparison.**

Table 4: **Ablation study on Tanks and Temples.**

| | GCR | DNC | PCR | GD | $F_1$ ↑ | PSNR ↑ | SSIM ↑ | LPIPS ↓ |
|---|---|---|---|---|---|---|---|---|
| Baseline | | | | | 0.07 | **25.11** | 0.851 | 0.179 |
| Model-A | ✓ | | | | 0.30 | 25.05 | 0.851 | 0.180 |
| Model-B | ✓ | ✓ | | | 0.32 | 24.98 | 0.849 | 0.184 |
| Model-C | ✓ | ✓ | ✓ | | 0.34 | 24.90 | 0.852 | 0.175 |
| Geo-3DGS | ✓ | ✓ | ✓ | ✓ | **0.36** | 24.93 | **0.854** | **0.170** |

**Depth-normal consistency (DNC).** By using depth-normal consistency introduced in Huang et al. (2024); Jiang et al. (2023), Model-B can further improve the reconstruction, but the rendering is negatively affected. This also explains why 2DGS degrades the rendering performance.

**Multi-view photometric consistency regularization (PCR).** By adding our proposed regularization, Model-C improves both reconstruction and rendering. This is because our regularization makes 3D Gaussians geometrically consistent across multiple views in our SDF initialization phase. This demonstrates the multi-view geometry consistency is beneficial to both rendering and reconstruction.

**Geometry assisted Gaussian densification (GD).** Our proposed multi-view photometric consistency is imposed through local image sampling. By adding the geometry assisted Gaussian densification, our Geo-3DGS achieves the best reconstruction and rendering performance. This is because our proposed geometry assisted Gaussian densificiation can perceive more global geometry information, encouraging the learned Gaussians to distributed more uniformly on the entire scene surface.

## 5 CONCLUSION

We have proposed Geo-3DGS, a new method to leverage multi-view geometry consistency for 3DGS and surface reconstruction. In our paper, we first reveal that there exists multi-view geometry inconsistency in 3DGS which prevents 3DGS from achieving high-quality rendering and reconstruction. Based on this insight, we develop a multi-view photometric consistency regularization to mitigate this inconsistency, thereby facilitating both rendering and reconstruction. Moreover, we design a geometry consistency-based SDF learning to extract meshes for 3DGS. This allows our method to exploit the geometric information derived from 3DGS for efficient SDF learning. In this way, our method achieves both high-quality rendering and accurate surface reconstruction in a unified framework for 3DGS while keeping a good efficiency.

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

# A APPENDIX

In this Appendix, we first discuss why we use SDF to learn surfaces for 3DGS A.1. Then, we provide more implementation details in Section A.2. In Section A.3 and Section A.4, we show more surface reconstruction and rendering comparisons on DTU and Tanks and Temples datasets. Next, we show the qualitative results of our ablation models in Section A.5. Finally, we discuss the limitation and future work of our method in Section A.6.

## A.1 WHY USING SDF TO LEARN SURFACES FOR 3DGS

The Gaussian points of 3DGS are discrete, sparse and far away from the surface. This makes directly extracting surface (e.g., Poisson surface reconstruction used in SuGaR) from these points very difficult. In contrast, SDF field is a continuous representation that can extract surface from the zero-level set. Therefore, we introduce the SDF field to learn continuous surfaces for 3DGS. In addition, recently, 2DGS introduces TSDF fusion to fuse depth maps rendered by 3DGS to reconstruct surface and achieves promising results. However, the used TSDF fusion is usually limited by fusion resolution. The SDF field can be flexibly used to extract surfaces with different resolutions. Also, the geometry constraints of 2DGS sacrifice rendering quality to some extent, as shown in Table 3. Therefore, we introduce the SDF field to decouple the geometry for 3DGS, avoiding affecting the rendering quality of 3DGS. This also makes our geometry can be used independently for downstream tasks. Moreover, the SDF field represented by MLP or hash grids is usually lightweight compared to the heavy 3DGS representation. Therefore, combining SDF with 3DGS is promising in practice.

## A.2 IMPLEMENTATION DETAILS

Our Geo-3DGS is implemented by PyTorch Paszke et al. (2019) with the Adam optimizer Kingma & Ba (2014). Our geometry network $f_\theta$ contains the hash grid Müller et al. (2022) with 16 levels and 2 feature channels per level, and 2-layer MLPs with a hidden size 64. The geometry network is initialized by geometric initialization Atzmon & Lipman (2020). Position encoding Mildenhall et al. (2020) is applied to query points. Following 3DGS Kerbl et al. (2023), we train our Geo-3DGS for 30k iterations. The multi-view photometric consistency regularization is imposed during 7k and 9k iterations. The geometry assisted Gaussian densification is applied at the 9k-th iteration. For our geometry consistency-based SDF learning, the 3DGS-driven SDF initialization is implemented during 9k to 12k iterations. Then, the geometry-consistent SDF refinement is applied for 3k iterations on DTU and 4k iterations on Tanks and Temples. We sample 4096 pixel rays in a mini-batch. In our loss function, all weights $\{\lambda_i\}_{i=1}^9$ are set the same for all scenes. Specifically, $\lambda_1 = 0.2$, $\lambda_2 = 0.5$, $\lambda_3 = 0.05$, $\lambda_4 = 0.01$, $\lambda_5 = 0.5$, $\lambda_6 = 0.25$, $\lambda_7 = 0.25$, $\lambda_8 = 1.0$, $\lambda_9 = 0.01$. Besides, in our multi-view geometry consistency check, $\epsilon_{\text{diff}}$ and $\epsilon_{\text{reproj}}$ are set to 0.01 and 1, respectively. After training, the triangle mesh is extracted from the geometry network by Marching Cube Lorensen & Cline (1987). The extraction resolution is set to $512^3$ for DTU and $2048^3$ for Tanks and Temples.

## A.3 MORE SURFACE RECONSTRUCTION COMPARISON

Figure 6 shows more surface reconstruction comparison on DTU. As can been seen, our method yields more complete and consistent surfaces than other SOTA methods. Figure 7 shows the comparison of reconstructed results on Tanks and Temples dataset in terms of the Precision metric. As shown, 3DGS Kerbl et al. (2023) suffers from poor geometric consistency of its rendered depth maps. SuGar Guédon & Lepetit (2023) demonstrates limited improvement in alleviating the geometric inconsistency of 3DGS especially in the texture-less areas. Although 2DGS can recover some details on the large-scale scenes of Tanks and Temples, it is still unable to reconstruct accurate surface meshes. Compared with these methods, our Geo-3DGS is not only able to recover the delicate details but also to improve the geometric consistency of the reconstructed surface.

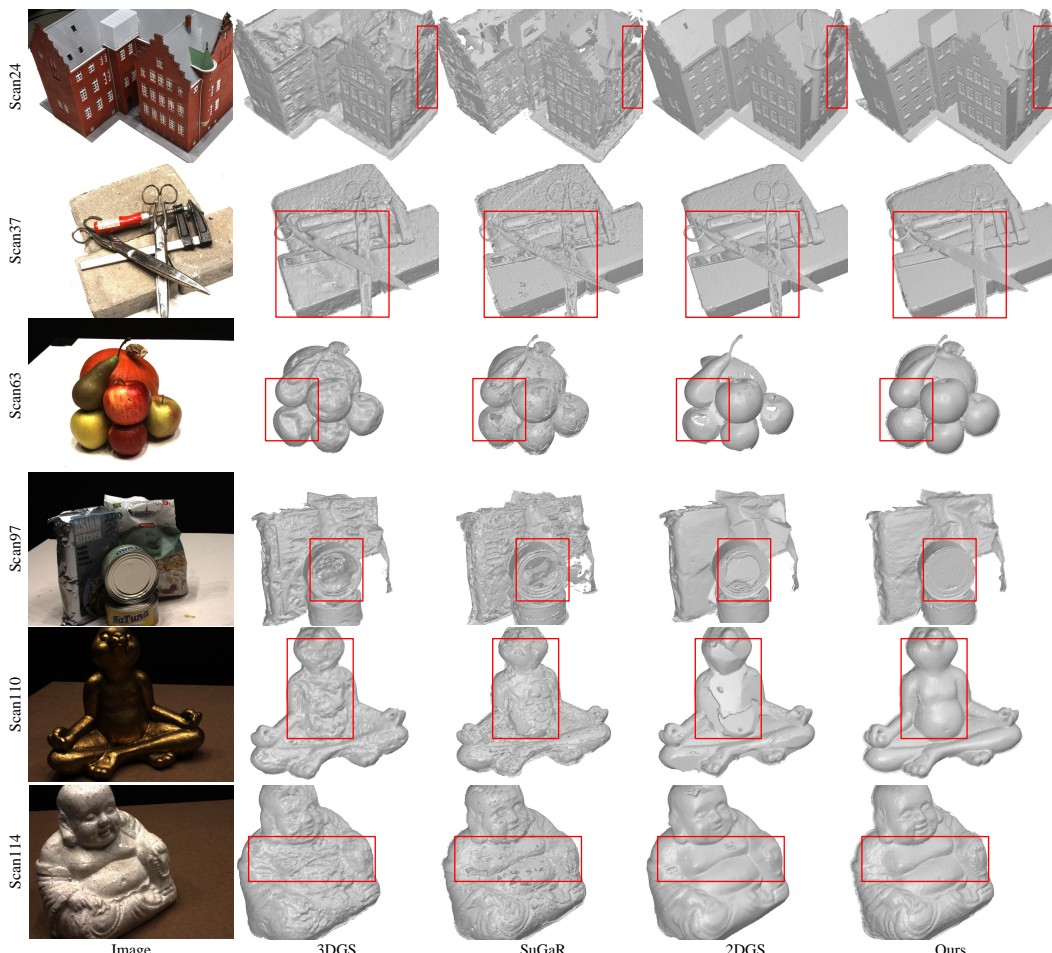

Figure 6: **More qualitative reconstruction comparison on DTU.**

## A.4 MORE RENDERING COMPARISON

We show additional qualitative rendering comparison for the DTU dataset in Figure 8. In Figure 9, we show rendering comparison results on more scenes from Tanks and Temples dataset. We observe that our method can render more visually realistic novel views than other methods, especially synthesizing more fine textures. This demonstrates the superior NVS performance of our proposed method.

## A.5 QUALITATIVE RENDERING AND RECONSTRUCTION COMPARISON OF ABLATION MODELS

Figure 10 shows the qualitative rendering comparison of ablation models. As can be seen, without our proposed geometry consistency strategies for 3DGS, the rendering results of Baseline, Model-A and Model-B are blurry in some regions with detailed textures, such as the glasses in the Barn scene of Figure 10. By employing our proposed multi-view photometric consistency regularization and geometry assisted Gaussian densification, Model-C and our full model can render these detailed textures much better. This demonstrates the effectiveness of these two strategies for NVS. In addition, Figure 11 presents qualitative comparison of Gaussian points and rendering results with Baseline model and ours Geo-3DGS. It can be seen that the proposed geometry assisted Gaussian densification can generate new Gaussians that are uniform distribution throughout the scene, resulting in more realistic rendering results. Figure 12 shows the qualitative reconstruction comparison of ablation models, which illustrates that the proposed strategies are beneficial for improving the fidelity of the reconstructed surface.

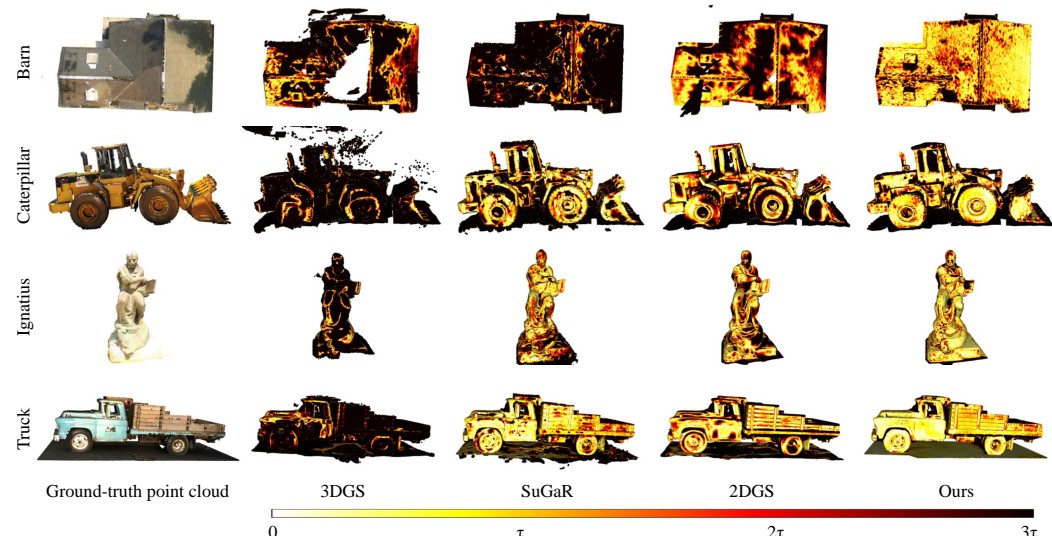

Figure 7: **Comparison of reconstructed results with state-of-the-art methods on Tanks and Temples dataset in terms of the Precision metric.** $\tau$ is the scene-relevant distance threshold determined officially and darker regions indicate larger error encountered with regard to $\tau$, which is set to 10mm, 5mm, 3mm and 5mm on Barn, Caterpillar, Ignatius and Truck, respectively.

Table 5: **Ablation study on Tanks and Temples.** GCR, DNC, PCR and GD denotes geometry-consistent SDF refinement, depth-normal consistency, multi-view photometric consistency regularization and geometry assisted Gaussian densification, respectively.

|  | $F_1 \uparrow$ | PSNR $\uparrow$ | SSIM $\uparrow$ | LPIPS $\downarrow$ |
| --- | --- | --- | --- | --- |
| w/o GCR | 0.29 | 24.70 | 0.851 | 0.175 |
| w/o DNC | 0.33 | **25.04** | **0.856** | **0.166** |
| w/o PCR | 0.33 | 24.98 | 0.848 | 0.174 |
| w/o GD | 0.34 | 24.90 | 0.852 | 0.175 |
| Full model (Geo-3DGS) | **0.36** | 24.93 | 0.854 | 0.170 |

In addition, we conduct another ablation study to show the effectiveness of the designs in our method. Specifically, we will remove one component from our full model at a time. The results in Table 5 show that each module improves surface reconstruction. As for rendering quality, the depth-normal consistency slightly affects the rendering quality. Overall, our introduced modules can better take both surface reconstruction and rendering into consideration.

To further show the effecacy of combining SDF with 3DGS, We utilize TSDF fusion based on our depth maps to test TNT dataset. The results in Table 6 show that our method achieves better reconstruction performance.

A.6 LIMITATION AND FUTURE WORK

Our proposed Geo-3DGS follows 3DGS to use spherical harmonics for view-dependent appearance modeling. This may limit its NVS performance for scenes with reflections. Compared with Neuralangelo Li et al. (2023), our method still has room for improvement in surface reconstruction. Incorporating the numerical gradients-based training and coarse-to-fine SDF training strategy from Neuralangelo into our method may further improve our surface reconstruction. In addition, it will be interesting to explore our proposed geometry consistency strategies based on 2DGS Huang et al. (2024).

Table 6: **TSDF fusion based on our detph maps.**

|                | Barn | Cat. | Court. | Ign. | Meet. | Truck | Mean |
|----------------|------|------|--------|------|-------|-------|------|
| TSDF           | 0.38 | 0.21 | 0.16   | 0.46 | 0.13  | 0.40  | 0.29 |
| Ours (Geo-3DGS)| 0.44 | 0.30 | 0.20   | 0.57 | 0.22  | 0.42  | 0.36 |

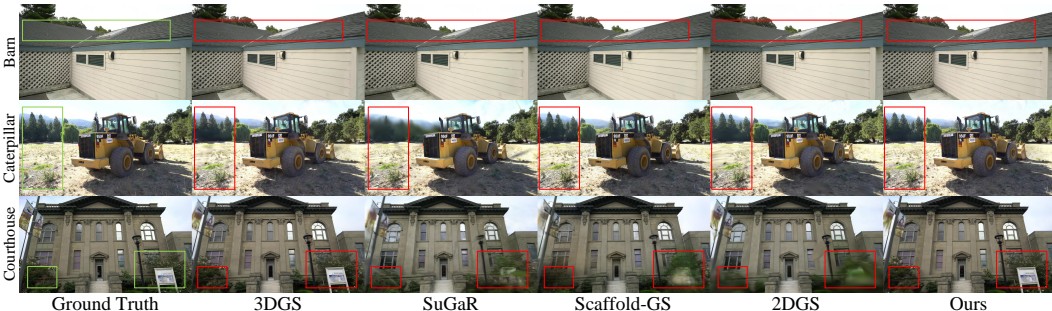

Figure 9: **More qualitative rendering comparison on Tanks and Temples.**

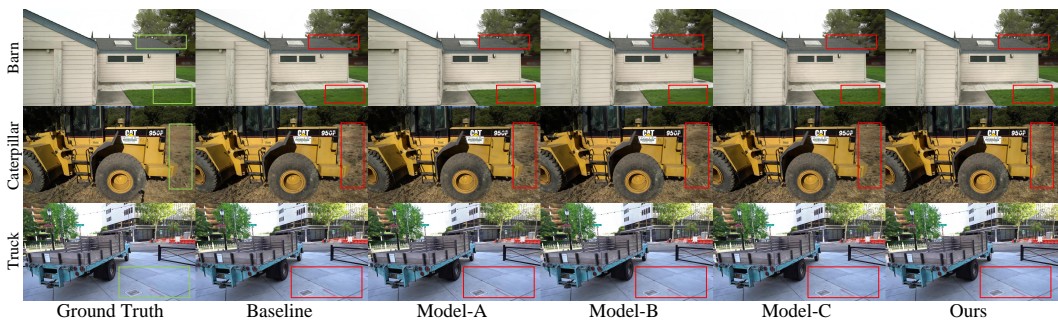

Figure 10: **Rendering quality of ablation models.**

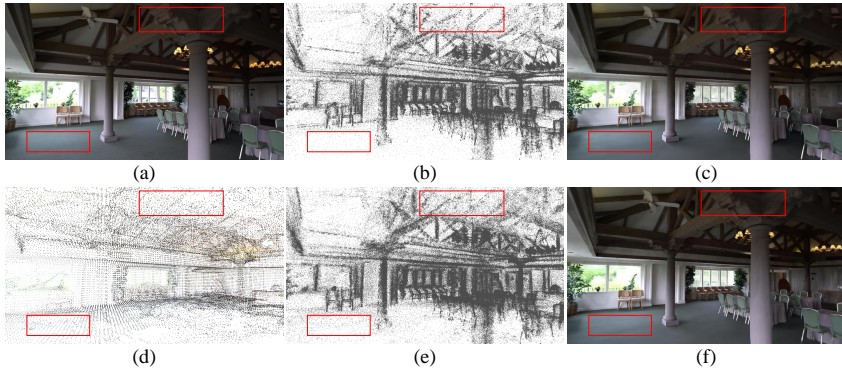

Figure 11: **Comparison of Gaussian points and rendering results with Baseline model and ours Geo-3DGS.** (a) is the ground truth image, (b) and (c) are the final Gaussian points and rendered image of Baseline, (d)-(f) are the new Gaussian points generated by geometry assisted Gaussian densification, final Gaussian points and rendered image of ours Geo-3DGS.

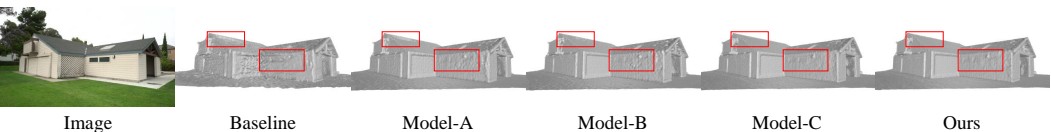

Figure 12: **Reconstructed meshes of ablation models.**

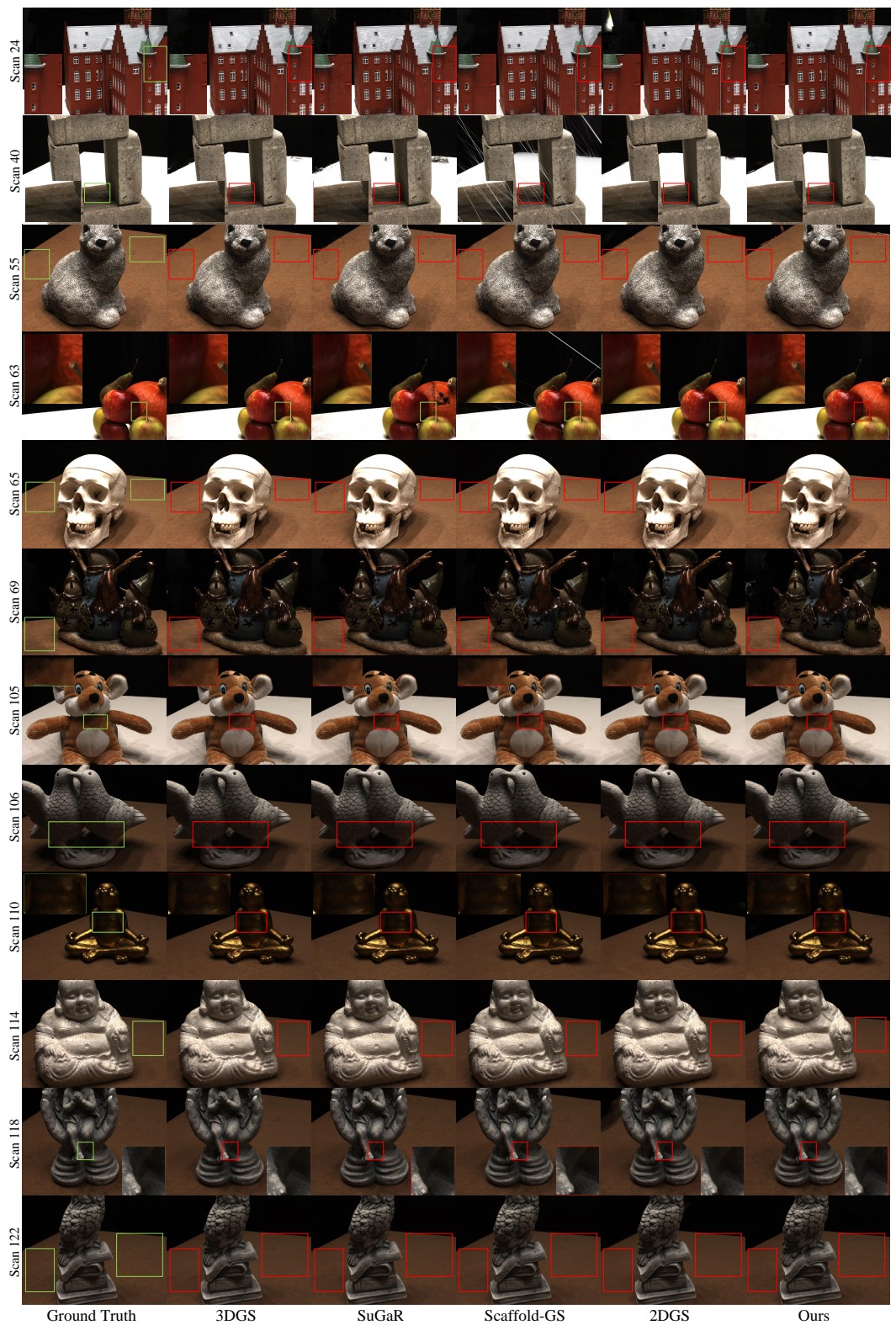

Figure 8: **More qualitative rendering comparison on DTU.**

