# OpenReview forum: "Geo-3DGS: Multi-view Geometry Consistency for 3D Gaussian Splatting and Surface Reconstruction"
_ICLR.cc/2025/Conference — Submitted to ICLR 2025_

### Official Review · Reviewer_St37 · 2024-10-27

**Soundness:** 3
**Presentation:** 3
**Contribution:** 2
**Rating:** 5
**Confidence:** 4

**Summary:**

Geo-3DGS first analyzes the consistency problem of depth maps generated from 3DGS and proposes a multi-view photometric consistency regularization to address it. Geo-3DGS utilizes two branches to model scenes: GS for rendering and SDF for surfaces, respectively. Comprehensive experiments demonstrate that Geo-3DGS achieves excellent performance.

**Strengths:**

1. Geo-3DGS reveals that the depth maps from the GS representation are not consistent across multiple-view images and introduces a regularization technique to enforce multi-view consistency.
2. The paper is good written and easy to follow.

**Weaknesses:**

1. **Lack of Novelty.** The concept of combining the advantages of 3DGS with neural SDF to enhance both rendering and reconstruction quality has already been proposed by GSDF. It would be beneficial to explain the differences in detail and conduct a comprehensive comparison experiment.
    1. In Section 'Geometry-Assisted Gaussian Densification,' how can the balance between new Gaussian primitives derived from depth maps and the existing Gaussians be achieved? Moreover, since depth maps generated by 3DGS are generally of lower quality compared to those from SDF, why not use depth maps from SDF to guide Gaussian densification, similar to the approach in Section 3.2.2 'Geometry-Aware Gaussian Density Control' in GSDF?
    2. In Section ‘3DGS-driven SDF initialization’, the idea of depth rendered from 3DGS to help initalize SDF is also similar with GSDF, in sec. 3.2.1 ‘Depth Guided Ray Sampling’.
2. **Poor typography.** In Tables 1 and 2, the timing for each baseline is completely incorrect. In Figure 5, it's unclear what's happening—there are too many lines cluttering the figures.
3. **Poor performance.** In surface reconstruction, Geo-3DGS performs significantly worse than Neural-Angelo in Table 2. I'd also like to see Neural-Angelo's performance on the DTU dataset in Table 1. In terms of rendering quality, Geo-3DGS still underperforms compared to 3DGS and Scaffold-3DGS, as shown in Table 3.
4. **Lack of significant improvement.** In the ablation study, the multi-view photometric consistency regularization, which is the paper's most novel contribution, only improves the F1 score by 0.02.
5. **No demos submitted**

**Questions:**

Which SDF branch did the authors use? Is it the hash-encoding-based Neus? What are the training iterations for the SDF branch? Also, why is the training time for Geo-3DGS only 48 minutes on the Tanks and Temples dataset and 45 minutes on the DTU dataset? I would expect training on Tanks and Temples to be much more challenging than training on DTU.

---

> ### Author Response · Authors · 2024-11-25
> **Response to Reviewer St37**
>
> We would like to sincerely thank the reviewer for the detailed and insightful comments on our work. We take every comment seriously and hope our response can address the reviewer’s concerns. If there are any remaining questions, we are more than happy to address them.
>
> **W1. Difference from GSDF.**
> GSDF directly use the NeuS framework to learn the SDF, which requires learning an implicit radiance field. Different from GSDF, our SDF learning does not rely on implicit radiance field learning and further leverages multi-view photometric consistency regularization. To balance new Gaussian primitives derived from depth maps and the existing Gaussians, we empirically employ downsampling to control the number of new Gaussians.
>
> For densification, GSDF first trains 3DGS branch for 15k iterations until convergence. Then, after it warm-ups NeuS with 3DGS for 5k iterations, it then uses the depth map from SDF to guide 3DGS. This process is complex and grealy depends on the learning of NeuS. To better leverage the geometry information of 3DGS to facilitate 3DGS itself, we impose the multi-view photometric consistency to boost the geometry consistency of the depth maps rendered by 3DGS, and then we can use the more accurate depth information to help 3DGS densification.
>
> For our 3DGS-driven SDF initialization, it is common to use detph prior guided near-surface sampling to help SDF learning. In this part, our focus is how to leverage the geometry information of 3DGS to directly initialize SDF. Therefore, we leverage rendered depths (Eq. (10)), zero-level set constraint (Eq. (10)), and weight distribution constraint (Eq. (11)) to achieve this, which are different from GSDF.
>
> **W2. Typography.**
> We are very sorry for these typos and will modify them in our revised version.
>
> **W3. Poor performance.**
> Although NeuralAngelo achieves the best performance on Tanks and Temples dataset, it requires a very long training time. Compared to the concurrent GSDF, GDSR that also leverages SDF, our performance is much better than these methods. Compared to 3DGS and Scaffold-GS, our achieved LPIPS is much better than theirs. Moreover, as shown in Figures 5, 8, 9, compared to 3DGS and Scaffold-GS, our method can significantly reduce blurring artifacts thanks to our geometry-assisted Gaussian densification.
>
> **W4. Lack of significant improvement for the multi-view photometric consistency regularization.**
> How to combine 3DGS and SDF to improve both rendering and surface reconstruction is still an open problem. To achieve this goal, our proposed method is a systematic design. For the improvement of the multi-view photometric consistency regularization, although it improves surface reconstruction by 0.02, it improves the rendering metric, LPIPS by 0.009, which significantly reduces artifacts as shown in Figure 10.
>
> **W5. Demo.**
> We have submitted reconstructed surfaces demos in our supplementary.
>
> **Q1. SDF architecture.**
> We use the hash-encoding-based SDF. Note that, we do not introduce implicit color fields like NeuS, as mentioned in L351. We follow 2DGS to downsample images by a factor of 2 (978x546) for Tanks and Temples dataset, and use full-resolution images (1600x1200) for DTU dataset. The downing operation accelerates the convergence on Tanks and Temples dataset.

---

> > ### Comment · Reviewer_St37 · 2024-11-25
> >
> > **performance**
> >
> > 1. "Although it improves surface reconstruction by 0.02, it only improves the rendering metric, LPIPS, by 0.009." I believe an LPIPS improvement of 0.009 is insufficient to demonstrate a meaningful module.
> > 2. "Compared to 3DGS and Scaffold-GS, our LPIPS performance is significantly better." However, as shown in Table 3, both 3DGS and Scaffold-GS achieved better visual quality than Geo-3DGS.
> > 3. "Compared to the concurrent GSDF, our performance is much better." Could you provide some qualitative results to support this claim? I believe GSDF demonstrates both high-fidelity surface reconstruction and excellent rendering quality.
> >
> > **model design**
> >
> > 1. GSDF does not rely on implicit radiance field learning. Instead, it interpolates information from 3DGS and achieves better results.
> > 2. For densification, GSDF leverages SDF with a novel grow-and-prune operator to guide GS primitives closer to the surface, rather than using depth maps from SDF.
> > 3. The idea of using depth-prior-guided near-surface sampling to assist SDF learning is similar to GSDF. Although the implementation differs slightly, both methods utilize the geometric information from 3DGS to achieve better SDF initialization. Additionally, both approaches recognize the limitations of 3DGS accuracy and provide corresponding solutions.

---

### Official Review · Reviewer_MRvc · 2024-10-27

**Soundness:** 3
**Presentation:** 3
**Contribution:** 2
**Rating:** 5
**Confidence:** 4

**Summary:**

This paper aims to improve the quality of the rendering results and learned geometry. They exploited the multi-view projection to check the inconsistency of 3DGS and used patch-based multi-view projection to add consistency regularization. Meanwhile, they leveraged these consistency information to guide the densification of 3DGS, combined the SDF module to learn the continue geometry, and injected the multi-view projection to constrain the SDF module.

**Strengths:**

This paper investigate the inconsistency in 3DGS and incorporate the multi-view projection to assist the regularization, densification and  geometry optimization of 3DGS model:
- use the multi-view projection to achieve the consistency regularization.
- use the filtered multi-view consistent depth to achieve densification of Gaussian points.
- use the SDF module to optimize the geometry and inject the multi-view projection.

**Weaknesses:**

- to my knowledge, multi-view projection is also a strategy to enhance the accurate of geometry in the multi-view visible region, and it still cannot mitigate the small-overlap or textureless issues you mentioned. And the example shown in Fig. 1 is still a texture-rich scenes (not the true textureless scene like the white background) to some extent, thus the depth of the table region can still be retained.
- maybe combining the SDF module with the 3DGS cannot be regarded as a contribution of this paper, as there are some methods have tried this strategy like NeuSG [1], GSDF [2] and 3DGSR [3]. Maybe the difference is that this paper further use the geometry constraints proposed in Geo-NeuS [4], and I think it's hard to say that this module is novel enough.
- the ablation and experiment of the densification strategy are not enough. This paper used the downsample to control the densified points, and how the number of points affect the performance and efficiency?
- some typos, like Line 261, Tab. 1 and Tab. 2.


[1] NeuSG: Neural Implicit Surface Reconstruction with 3D Gaussian Splatting Guidance, arxiv 2023.
[2] Gsdf: 3dgs meets sdf for improved rendering and reconstruction, arxiv 2024.
[3] 3dgsr: Implicit surface reconstruction with 3d gaussian splatting, arxiv 2024.
[4] Geo-neus: Geometry-consistent neural implicit surfaces learning for multi-view reconstruction.

**Questions:**

In the multi-view photometric consistency regularization, how do you choose the base pixel used to calculate the multi-view consistency? And the 3DGS is not the ray-based rendering model like NeRF or NeuS, why do you still choose pixel/patch based projection instead of directly using the entire rendered depth to compute the multi-view projection loss? So what about the performance differences between these two strategies?

---

> ### Author Response · Authors · 2024-11-25
> **Response to Reviewer MRvc**
>
> We would like to sincerely thank the reviewer for the detailed and insightful comments on our work. We take every comment seriously and hope our response can address the reviewer’s concerns. If there are any remaining questions, we are more than happy to address them.
>
> **W1. Small-overlap or textureless issues.**
> Traditional photometric consistency cannot tackle these issues well based on the multi-view projection. However, as demonstrated in [1] [2], computing photometric consistency using multi-view deep features can greatly overcome these issues. Therefore, we think that the introduced multi-view photometric consistency is still an important cue for 3DGS.
>
> [1] Learning signed distance field for multi-view surface reconstruction, CVPR 2021.
>
> [2] PSDF: prior-driven neural implicit surface learning for multi-view reconstruction, TVCG 2024.
>
> **W2. Difference from NeuSG, GSDF and 3DGSR.**
> Both NeuSG and GSDF introduce an extra implicit radiance field to help SDF learning. Unlike them, our method directly learns SDF from the geometry of 3DGS. 3DGSR learns the normal information of 3DGS by introducing a new attribute for each Gaussian by following R3DG [3]. This cannot effectively guarantee the intrinsic geometry-normal consistency of 3DGS. Different from these methods, we reveal that the multi-view geometry consistency is also important for 3DGS and leverage this consistency to further facilitate 3DGS and more efficient SDF learning.
>
> [3] Relightable 3D Gaussians: Realistic Point Cloud Relighting with BRDF Decomposition and Ray Tracing, ECCV 2024.
>
> **W3. Some typos.**
> We are very sorry for these typos and will further modify them in our revised version.
>
> **Q1. How to choose base pixels?**
> We randomly choose base pixels. We do not directly use the entire rendered depth to compute the multi-view photometric consistency regularization because it will introduce expensive computation cost for 3DGS. Therefore, we still employ pixel/patch based projection.

---

> > ### Comment · Reviewer_MRvc · 2024-11-25
> >
> > Thanks for authors' response, but it doesn't solve all concerns appropriately. In my opinion, using depth-based multi-view photometric consistency will not introduce more efficiency problem than the pixel-based consistency method, and maybe show some relative comparisons will be more convincing. And this explanation also don't solve the question about the performance difference. Meanwhile, in the SDF branch, the design is too similar to Geo-NeuS, and perhaps the author should consider more diverse contributions.

---

### Official Review · Reviewer_WBQT · 2024-11-04

**Soundness:** 3
**Presentation:** 3
**Contribution:** 2
**Rating:** 5
**Confidence:** 4

**Summary:**

The paper introduces a method to regularize the surface reconstruction process in Gaussian Splatting while preserving its rendering quality. Although Gaussian Splatting is effective for rendering, it has limitations regarding multi-view consistency, which this paper addresses. As illustrated in Figure 2, these limitations result in poor depth maps, leading to suboptimal 3D surface reconstructions. To address these issues, the authors propose integrating traditional multi-view stereo techniques, such as PatchMatch [1], to enhance multi-view consistency within the Gaussian Splatting framework. The paper's core contribution is to adapt and extend these techniques to improve surface reconstruction quality while preserving Gaussian Splatting’s high rendering fidelity. Experimental results on the Tanks and Temples and DTU datasets demonstrate that this approach achieves improved surface reconstruction quality without compromising rendering performance.

[1] Schönberger, Johannes L., et al. "Pixelwise view selection for unstructured multi-view stereo.", ECCV 2016.

**Strengths:**

**Clarity**: The paper is well written and the entire math is easy to follow.\
**Reproducibility**: The code will be released if accepted which will help to reproduce results. \
**Significance**: Extracting high quality 3D surface from Gaussian Splatting without affecting the rendering quality is a relevant problem.

**Weaknesses:**

**Novelty:** While the paper presents promising results on the surface reconstruction benchmark, the contribution feels somewhat incremental. The approach primarily extends regularization techniques from NeRF-based methods, such as those introduced in GeoNeus [2], to the Gaussian Splatting framework. The equations used closely resemble those in GeoNeus, with both approaches aiming to improve surface consistency, though GeoNeus achieves this within a volume rendering framework, whereas this work applies it to Gaussian Splatting. It would be helpful if the authors could further clarify the distinct advantages of adapting these regularization specifically to Gaussian Splatting, as well as any unique challenges addressed in this framework.

**Results:** Although this paper shares similarities in approach with GeoNeus, it does not provide a direct comparison of surface reconstruction quality with GeoNeus, as seen in Table 1 for the DTU dataset. Additionally, the improvements in surface reconstruction quality over previous methods appear modest. In the Tanks and Temples dataset, the method’s reconstruction quality is notably lower than that of Neuralangelo [3]. Clarifying the reasons for these results, particularly regarding the performance gap with Neuralangelo, would strengthen the evaluation and highlight any inherent trade-offs or challenges in adapting this approach within the Gaussian Splatting framework.

[2] Fu, Qiancheng, et al. "Geo-neus: Geometry-consistent neural implicit surfaces learning for multi-view reconstruction.", NeurIPS 2022. \
[3] Li, Zhaoshuo, et al. "Neuralangelo: High-fidelity neural surface reconstruction." CVPR, 2023.

**Questions:**

- The convergence time of Neuralangelo seems significantly higher (24hr) compared to this method (48m) as shown in table 2. What is the reason behind this even though both the methods use hash grids? A plot regarding time vs chamfer distance or time vs F1 score will help to understand the convergence speed of both the methods correctly.

- I am not quite sure what is the purpose of Equation 11? An ablation regarding its effect on the final reconstruction accuracy will be helpful.

---

> ### Author Response · Authors · 2024-11-25
> **Response to Reviewer WBQT**
>
> We would like to sincerely thank the reviewer for the detailed and insightful comments on our work. We take every comment seriously and hope our response can address the reviewer’s concerns. If there are any remaining questions, we are more than happy to address them.
>
> **W1. Distinct advantages of adapting photometric regularization to Gaussian Splatting and the unique challenges.**
> 3DGS can quickly converge and render images and depths in a real-time manner. In contrast, NeRF-based methods cannot render images and depths in a real-time manner. With the efficiently rendered depths by GS, our method can efficiently impose photometric consistency. However, only using this photometric regularization cannot guarantee satisfactory surface reconstruction and rendering because 1) Although photometric regularization improves the geometry consistency for 3DGS, how to reconstruct surfaces for 3DGS remains open. 2DGS fuse depth maps by TSDF to obtain surface models, which usually needs to tune hyper-parameters for TSDF and requires a large amount of memory. Some methods, such as Neusg and GSDF, use SDF to learn surfaces for 3DGS, but hey require learning implicit radiance field at the same time. Therefore, adapting photometric consistency to GS and learn surfaces with lightweight SDF is challenging . To circumvent memory-expensive TSDF fusion and time-consuming implicit radiance field learning, we leverages SDF and further introduce 3DGS-driven SDF initialization. With our designs, our method only uses the depth being optimized by 3DGS to efficiently guide SDF learning.
>
> **W2. Comparisons with GeoNeus and Neuralangelo.**
> We will add the evaluation results of GeoNeus on DTU. For Neuralangelo, it requires 8 GPUs and a very long training time to sample a large number of rays to get the results. In contrast, our method is efficient by training around 48 minutes on a single GPU. We think that an efficient reconstruction method is more practical in real-world applications.
>
> **Q1. Convergence time comparison with Neuralangelo.**
> Our method is more efficient because: 1) Neuralangelo requires learning implicit radiance filed, which introduces a huge number of extra hash grid evaluations. 2) Our method leverages explicit geometry cues from 3DGS. This can efficiently make SDF focus on near-surface learning.
>
> **Q2. Purpose of Eq. (11).**
> This equation aims to constrain the weight distribution of SDF, regularizing the noise far from the depths of 3DGS.

---

### Official Review · Reviewer_6fqk · 2024-11-04

**Soundness:** 3
**Presentation:** 4
**Contribution:** 2
**Rating:** 5
**Confidence:** 4

**Summary:**

The paper presents a new 3D reconstruction method that extends 3D Gaussian Splatting. In detail, the paper presents two main contributions on top of 3D GS.
1. Usage of photometric loss as a way to regularize ambiguity, augment densification
2. Usage of SDF as a way to promote geometric consistency
In addition to these contributions, the paper provides evaluation on multiple standard dataset for benchmarking against existing state of the art approaches.

**Strengths:**

The paper makes it clear that the additional regularizations, derived from insights from existing 3D reconstruction methods that promote joint photometric / geometric regularization such as COLMAP MVS, vastly helps the quality and reliability of reconstruction. Each contributions are tailored for 3D Gaussian Splatting pipeline. For example, photometric loss is not only used as a regularization (which is commonly used in existing NeRF or MVS literatures), but as a densification. Similarly, SDF is not used as a geometric representation, but rather, as a way to promote geometric consistency.

**Weaknesses:**

Major concern with the paper is that the method is not new. Usage of photometric consistency as well as geometric consistency has been around for years in 3D reconstruction, starting from MVS to NeRF. The reviewer understands that these may not have been brought in, in the domain of 3D GS, but finds it unsure if the contributions are enough for ICLR.

Also, based on the ablation studies provided in the table 4, the reviewer finds photometric regularization and densification to be not as important, compared to the SDF based geometric consistency. It seems like that the model is going for rendering vs. geometric accuracy tradeoff, which is hard to justify as major contribution.

**Questions:**

What types of SDF architecture has been used? There has been advancements in SDF networks that may benefit your model as well.

Has the authors tried different initialization schemes? There may be better way to initialize SDF given that 3D Gaussians reside near the surfaces (e.g, solving Poisson Equation using Gaussin Splats)

---

> ### Author Response · Authors · 2024-11-25
> **Response to Reviewer 6fqk**
>
> We would like to sincerely thank the reviewer for the detailed and insightful comments on our work. We take every comment seriously and hope our response can address the reviewer’s concerns. If there are any remaining questions, we are more than happy to address them.
>
> **W. Difference from MVS and NeRF, and our contributions.**
> 1) *Efficient and accurate surface reconstruction with 3DGS.* MVS usually adopts depth map estimation and fusion to obtain point clouds and then obtain surfaces by Poisson surface reconstruction. NeRF uses SDF to learn neural implicit surfaces, which usually requires simultaneous implicit radiance field learning. Different from these works, our introduced SDF directly learns neural surfaces from the geometry of 3DGS without the need to learn the implicit radiance field. To effectively achieve this, we find that the multi-view geometry consistency is ignored by 3DGS, which is never revealed by previous works. Therefore, we leverage the photometric consistency to encourage multi-view geometry consistency.
>
> 2) *Photometric regularization and densification help both reconstruction and rendering.* Photometric regularization facilitates the geometry consistency of 3DGS. This improves the SDF initialization efficiently, which is hard for previous NeRF-based methods. On the other hand, photometric regularization helps populate empty areas for 3DGS, as shown in Figure 1(a). Moreover, since 3DGS gradually densifies from sparse point clouds computed by SfM, it is usually challenging for 3DGS to reconstruction the entire 3D scene completely. Our densification leverages more complete geometry information to encourage 3D Gaussians densification. By comparing Model-B and Model-C, Model-C and Geo-3DGS in terms of F1 and LPIPS metrics, our designs improve both reconstruction and rendering. The ablation results in Figures 10 and 12 further support this.
>
> **Q1. SDF architecture.**
> In L731, we show that our SDF architecture is based on hash grids.
>
> **Q2. SDF initialization.**
> Following the practices in NeRF-based surface reconstruction methods, we use the geometric initialization (Atzmon\&Lipman(2020)) to initialize our SDF for fair comparisons with these methods. Solving Poisson Equation using Gaussian Splats to initialize SDF is interesting. On the other hand, as the Gaussian splats of 3DGS are inconsistent geometry information, we believe that photometric consistency still can help the SDF initialization.

---

> > ### Comment · Reviewer_6fqk · 2024-11-27
> >
> > Thank you authors for the feedback on the comments.
> > The reviewer still finds the claim on "To effectively achieve this, we find that the multi-view geometry consistency is ignored by 3DGS, which is never revealed by previous works" to be not as clear. If the reviewer understands correctly, the 3D GS optimization process is driven by multi-view consistent renderings as a loss function.
> >
> > Next, the reviewer still finds it uncanny whether the claim "Different from these works, our introduced SDF directly learns neural surfaces from the geometry of 3DGS without the need to learn the implicit radiance field." is important. Why is this useful? If it is just to promote geometric consistency, the reviewer does not find enough evidence that the joint learning of SDF is the main contribution to achieving this. It could have been consistency of rendered depth (as COLMAP) or arbitrary sampled view (as RegNeRF)

---

### Comment · Area_Chair_Et4M · 2024-11-21
**Please initiate discussions!**

Dear authors and reviewers,

The discussion phase has already started. You are highly encouraged to engage in interactive discussions (instead of a single-sided rebuttal) before November 26. Please exchange your thoughts on the submission and reviews at your earliest convenience.

Thank you,
ICLR 2025 AC

---

### Comment · Area_Chair_Et4M · 2024-11-25
**Last day for interactive discussions!**

Dear authors and reviewers,

The interactive discussion phase will end in one day (November 26). Please read the authors' responses and the reviewers' feedback carefully and exchange your thoughts at your earliest convenience. This would be your last chance to be able to clarify any potential confusion.

Thank you,
ICLR 2025 AC

---

### Meta-Review · Area_Chair_Et4M · 2024-12-19

**Metareview:**

The submission received mostly slightly negative reviews. The reviewers' main concerns were on the similarity to prior works, lack of comparisons to critical baseline methods, and limited analyses. The AC carefully read through the paper, the reviewers' comments, and the authors' rebuttal. The AC agrees with the reviewers that the methods and analyses introduced in this paper do not justify for sufficient contributions. As such, the AC recommends rejection.

**Additional Comments On Reviewer Discussion:**

The reviewers raised questions mostly regarding lack of originality (6fqk, WBQT, MRvc, St37), insufficient comparative and ablative analysis (6fqk, MRvc, St37), and limited improvement in quality (WBQT, St37). The questions were addressed by the authors, but the reviewers were not convinced. The AC agrees with the reviewers' evaluation.

---

### Decision · Program_Chairs · 2025-01-22

Reject